# Enhanced-Precision Measurement of Glutathionyl Hemoglobin by MALDI-ToF MS

**DOI:** 10.3390/molecules28020497

**Published:** 2023-01-04

**Authors:** Federico Maria Rubino, Sara Ottolenghi, Andrea Brizzolari, Claudio Maioli, Michele Samaja, Rita Paroni

**Affiliations:** 1Laboratory for Analytical Toxicology and Metabonomics, Department of Health Sciences, Università degli Studi di Milano, v. A. di Rudinì 8, 20142 Milan, Italy; 2Department of Health Sciences, Università degli Studi di Milano, 20146 Milan, Italy; 3DAN Europe Research Division, 64026 Roseto degli Abruzzi, Italy; 4MAGI Group, 25010 San Felice del Benaco, Italy; 5Laboratory for Medical Mass Spectrometry, Department of Health Sciences, Università degli Studi di Milano, 20146 Milan, Italy

**Keywords:** glutathione, hemoglobin, method optimization, oxidative stress, red blood cells, MALDI-ToF, spreadsheet

## Abstract

Glutathionyl-hemoglobin (HbSSG) is used as a human biomarker to pinpoint systemic oxidative stress caused by various pathological conditions, noxious lifestyles, and exposure to drugs and environmental or workplace toxicants. Measurement by MALDI mass spectrometry is most frequently used, however, the method suffers from excessive uncontrolled variability. This article describes the improvement of a MALDI-ToF mass spectrometry method for HbSSG measurement through enhanced precision, based on strict control of sample preparation steps and spreadsheet-based data analysis. This improved method displays enhanced precision in the analysis of several hundred samples deriving from studies in different classes of healthy and diseased human subjects. Levels span from 0.5% (lower limit of detection) up to 30%, measured with a precision (as SE%) < 0.5%. We optimized this global procedure to improve data quality and to enable the Operator to work with a reduced physical and psychological strain. Application of this method, for which full instruction and the data analysis spreadsheet are supplied, can encourage the exploitation of HbSSG to study human oxidative stress in a variety of pathological and living conditions and to rationally test the efficacy of antioxidant measures and treatments in the frame of health promotion.

## 1. Introduction

The interplay of free sulfide (-SH) and disulfide (-S-S-) bonds in soluble thiols and in proteins is a prominent redox-sensitive process and has long been proposed as a clinical marker of oxidative stress [1].

Glutathionyl hemoglobin (HbSSG) is a minor form of human hemoglobin where glutathione (GSH = γECG; γ-glutamyl-cysteinyl-glycine) is covalently bound to the thiol group of the 93-cysteine amino acid residue of one β-chain in the protein tetramer through a disulfide (–S-S-) covalent bond (Figure 1). 

This bond can be generated also in the reaction of a free thiol with a thiol-sulfinic acid group, according to the scheme of Figure 2. The sulphinic acid can be either in the molecule of hemoglobin (pathway ***a***) or in that of glutathione (pathway ***b***) [2]. 

Vice-versa, the disulfide bond can be reversibly cleaved in a trans-sulfuration reaction [3] with a nucleophilic free thiol group that belongs to a compound with a lower redox potential [4,5]. These good reducing agent are in-vivo glutathione (pathway ***a***), drugs such as *N*-acetyl-cysteine and in-vitro glutathione, *N*-acetyl-cysteine and biochemical tools, such as dithio-treitol. 

Matrix-Assisted Laser Desorption Ionization mass spectrometry has been often used to measure HbSSG in the hemolysates of red blood cells (RBCs) with a fast turnaround time and minimal cost [6,7,8]. However, the reported methods did not address analytical quality issues in full, thus hampering a wider application of this biomarker in population studies.

As long known, the main weakness of MALDI-based methods for quantitative analysis of proteins and large peptides is the intrinsic, large shot-to-shot variability of relative signal intensities in the measurement of an individual sample. Spectra summation is used to decrease shot-to-shot variability to obtain a sample-representative spectrum, and meticulous sample preparation and deposition techniques are used to minimize heterogeneity in individual spots and between samples in the same analytical series. 

To understand how delicate is using MALDI-ToF for the quantitative assessment of biologically active proteins, a recently published protocol for the measurement of ubiquitin, a 8.5-kDa protein, in drug development studies resorts to use the biosynthetically-produced ^15^N-labelled protein as internal standard for quantification. The published method gives multiple warnings on critical steps throughout the procedure and on the several sources of possible error [9]. Other recent studies address the quantification in human blood of peptides and proteins, such as a phosphorylated amyloid peptide [10], monoclonal free light chains [11], C-peptide [12], isoforms of glutathione-S-transferase [13] and glycated hemoglobin [14,15].

A further source of method weakness derives from the unexpected inadequacy of spectra-processing software embedded in commercial instrument platforms [13]. Chromatographic software has been long developed to improve the accuracy and reproducibility of peak recognition and integration, especially with reference to peak-fitting weak and transient signals in fast chromatographic separations coupled to multi-experiment mass spectrometry detection. On the contrary, non-chromatographic mass spectrometry recordings yield a single spectrum, for which instrumental software often does not allow accurate integration of mass peaks. Quantitative applications need the highest level of inter-sample reproducibility in the integration phase, where the digitized spectrum profile is transformed in a series of molecule signals, each characterized by its mass-to-charge distinctive value and intensity as integrated peak area. Baseline smoothing and peak-searching algorithms are embedded in the data-processing software elaborated by instrument vendors, and account for most of the post-processing abilities available to the analyst to ensure measurement quality. However, in some cases, such as in our hands, for the measurement of HbSSG in human blood hemolysates, software capabilities could not ensure sufficient sample-to-sample homogeneity in replicates, thus artifactually decreasing the precision of the measurement, and downplaying the very promising use of this biomarker for clinically-oriented studies.

To improve measurement precision, we optimized several critical steps, from sample processing to spectrum integration in a method that had been developed and used in previous studies [7,8]. A breakthrough in this development was choosing not to use the vendor-supplied spectrum integration tools, and to develop an original, specific, totally operator-controlled spreadsheet calculation tool. We supply the spreadsheet as Microsoft Excel file in the Appendix A.

## 2. Results

The aim of this optimization study was to improve the precision of glutathionyl-hemoglobin measurement by MALDI-ToF in a standard MALDI-ToF instrument that does not have capacity for interfacing automated sample preparation devices. The first reported article [6] stated precision (as CV%) in the range of 4.6–7.5% for samples containing 4–8% HbSSG; however we could not consistently replicate the reported measurement quality on the same instrument, even by strictly following the reported procedure [7,8].

As an example, Figure 3 shows results from the quadruplicate measurement of a hemolysate sample that we selected among more than 300 similar ones only for sake of clear view of peak intensities, due to its high content of HbSSG. The four traces derive from four distinct depositions in consecutive sample plate spots from the same hemolysate-matrix mixture (instrumental replicates), measured with the automated acquisition method (FlexControl™ software).

Spectra are presented as displayed by the instrument’s FlexAnalysis™ software and were processed post-acquisition under the best-established conditions for HbSSG measurement. As apparent, the intensity of the individual spectra differ by a factor of approx. 2, even using the automated acquisition technique, and the β/α ratio of the intensities of the two chains of hemoglobin varies from 0.73 to 0.90%.

HbSSG is defined operationally as the ratio of HbSSG to the sum of β-Hb and of the glycated form (this minor form displays often as a badly resolved signal in the spectrum, due to the overlap of a matrix adduct peak). When the standard integration software is used for the displayed example measurements (the same sample measured in quadruplicate) this ratio varies from 13.1% to 14.5% (CV% = 4.7%), which is often an excessive variability for use as a clinical biomarker of oxidative stress. Peak integration is very variable, and unmanageable by the software parameters, even for the intense and well-resolved peak of β-Hb. The large variation of peak areas thus hampers a reproducible calculation of their ratio. As unavoidable consequence, the application of glutathionyl hemoglobin as a quantitative biomarker of oxidative stress is severely impaired.

To improve the quality of measurements, therefore, we optimized sample preparation and deposition, operative and instrumental parameters, including laser firing and mass spectrometer focusing, automation of measurement and integration, taking as the starting point those reported in our previous work. This optimization task was necessary since, due to the large number of samples that need to be measured in clinically oriented studies, measurements should be made unbiased and easy to perform by all involved operators. 

The most critical step towards enhancing precision proved to be post-acquisition integration of mass peaks with the FlexAnalysis™ proprietary software. Since we could not solve this problem by manipulating the available integration parameters in batch processing, we pursued an independent solution with the development and use of a custom spreadsheet. Once the integration bias was solved, the individual phases of sample preparation and instrumental analysis could be examined stepwise, until an optimized procedure was developed.

Optimization involved adaptation of some general instrumental procedures, preparation of red blood cell samples, sample preparation for MALDI analysis, sample deposition by manual “spotting”, spot analysis by MALDI laser rastering, post-acquisition signal integration and calculation of results.

### 2.1. Fine-Tuning of General Instrumental Routine Procedures

At the beginning of each measurement session of glutathionyl-hemoglobin, internal calibration of the *m/z* scale is accomplished in a random sample by using the signals of the main proteins in the hemolysate, homozygous α- and β-hemoglobin, present as the doubly- and singly- protonated ionic species (*m/z* 7564.2; 7934.2; 15,128.4 and 15,867.5 respectively). Table 1 displays the calculated elemental composition of the proteins and the corresponding *m/z* of the ionic species that appear in the MALDI analysis. Depending on the development phase, either a commercial preparation of human hemoglobin or a hemolysate of red blood cells were used, without any substantial difference in results.

Laser firing power is routinely adjusted at a sufficient value (percent of maximum allowable power) in order that peak intensity of the α-Hb chain (the most intense one) would be between 500–1000 cps in successful individual shots, with a final height of approx. 2500–5000 cps. This adjustment (35–50%) proved to be very stable over time and did not need frequent adjustments over more than two years of method operation. 

A final test of instrument performance is performed before starting each sample batch, by subjecting a random sample spot to the automated analysis procedure. After each laser shot, detection of a signal of α-Hb higher than 500 cps prompts the FlexControl™ software to add the raw spectrum to the integration register, until a number of 50 valid spectra are obtained on the sample spot and analysis is considered complete. The obtained spectrum is used to update mass calibration for the corresponding analysis batch.

### 2.2. Fine-Tuning of the Sample Loading Procedure

Sample loading (“*spotting*”) in MALDI analyses is the most critical, quality-determining physical step of the whole procedure. Due to the large number of samples measured in clinically oriented studies, manual sample preparation and loading is the most time-consuming, demanding, and critical procedure of MALDI analyses. Method optimization thus considers several ergonomic factors adapted to individual needs, aimed at avoiding unnecessary physical and psychological strain that causes errors due to repetitive movements [16] and loss of Operator concentration. To minimize strain-derived errors, we decided to limit a single batch to 24 samples, each loaded in quadruplicate in the 24 × 16 (396-position) sample plate of the instrument. This is the number of samples that a single, unaided Operator can safely manage within a two-day operating shift, without danger of sample loss or need for reanalysis.

When automated methods for coding/labeling, mixing and aliquoting the samples are not available, the most demanding phase of multiple sample preparation procedures is maintaining sample identity through the multiple manual passages from blood fractionation to data analysis. The choice of brands for consumables and plasticware proved critical for unexpected, apparently trivial reasons, which however influenced the turnaround time of analyses and the correctness of a few steps. 

The MALDI-ToF method uses minuscule amounts of sample and correspondingly small-sized laboratory plasticware, where coding a large number of items is often an eye- and hand-straining task, prone to error. PCR-grade sample cones are produced with different transparency, fastness of felt-tip ink writing, and tightness of lid. Failure in any of these properties yielded difficulties due to poor mixing of sample to the matrix solution, loss of sample identification, and loss of sample due to accidental, unexpected opening of the tube.

Choice of the brand of pipet tips, especially of the 2-microL ones used for final sample deposition on the stainless steel spot is another crucial step. Different brands are more or less elastic and allow the adoption of different techniques to obtain a successful deposition [17]. Very rigid ones do not give sufficient sensorial feedback to distribute sample volume and induce crystallization by a gentle “scratching” of the tip on the stainless steel surface. Softer (usually longer) ones do not allow sufficient control of fine movements, with the risk of rubbing the substance outside the spot circle.

The brands of materials that we describe in the Appendix A yield, in our hands, adequate results and are commonly available from multiple commercial suppliers.

### 2.3. Rationalization of the Operative Phases

The operative phases include sample preparation, spot loading, uploading the working list in the instrument software, automated sample analysis, and data download from the instrument, as outlined in Figure 4. The individual operative phases are summarized in the Section 3. The Appendix A section describes the detailed step-by-step procedure as applied in our laboratory.

We have organized the three phases in such a way that the work can be paused without detrimental effects on results quality and turnaround time. While assistance from a co-operator may be beneficial and timesaving, especially in Phases 1 and 2, the protocol is optimized for execution by a single Operator who is in charge of all phases.

The first phase entails the preparation of a spectrophotometrically titrated red blood cell hemolysate at a constant Hb concentration of 10 microM from a batch of 24 pre-fractioned blood samples, followed by aliquoting in the sample mini-tubes ready for MALDI analysis. As many as three or four batches (total 72 and 96 samples) can be processed daily, even by a single Operator, if all necessary sample tubes are pre-labeled and samples can be stored in the cold (ice bath or +4°C refrigerator) while waiting for the next phase. This has been indeed attempted with a favorable outcome; however, in our experience, it proved to be a *tour-de-force* stress test of laboratory logistics, rather than optimal management of instrument and Operator’s time.

The second phase encompasses the MALDI analysis, and the Operator can accomplish the task alone with the use of the pre-printed worksheets for sample tracking and sample-ID uploading in the instrument’s software for automated analysis. Manual loading of multiple sample batches on the MALDI plate is feasible, however at the cost of some eye- and hand-strain. The employed instrument and its available software do not allow for uploading previously generated worklists as Excel spreadsheets, therefore adding a further error-prone step of data entry to the global procedure.

Analysis of a fully loaded 396-well sample plate (94 samples in quadruplicate) takes approx. 5–10 hours of instrument time, depending on how many laser shots “go wasted” to achieve the 50 “valid” ones on each spot. For organizational reasons, unattended overnight analysis is best suited, with the repetition of aborted measurements being re-tried in the morning.

Analysis batches are processed offline to save instrument time at the shared facility laboratory, with the use of the spreadsheet that we developed for this task. Since this is the longest phase and, in our situation, it cannot be automated or simplified, it constitutes the bottleneck of data production. The first phase of data analysis of a 24-sample batch (96 spectra) takes approx. 2 hours and processing a single sample to obtain the final results takes 10–15 min, comprehensive of data transfer and manual elaboration time.

### 2.4. Improvement of Results Precision with the Use of a Custom Spreadsheet

As apparent from observing the colored areas of the peaks in the four spectra of Figure 2, the baseline chosen by the vendor’s peak integration tool included extraneous portions of the signal in the integration. Therefore, the intensities and the corresponding peak intensity ratio resulted unreliable. Some examples in the literature use Excel spreadsheets to analyze chromatographic profiles imported as binary time-intensity files [18] that are under several regards similar to the mass-intensity files generated in the MALDI-ToF analysis of red blood cell hemolysates.

The aim of our spreadsheet is to analyze together the four mass-intensity profiles that derive from the automated analysis of four instrumental replicates of the same prepared hemolysate. A key factor in obtaining matching profiles of the mass spectra is normalizing peak heights in the region of interest to the same value. This procedure allows comparing the starting and ending extremes of the “Gaussian” profile to select the same *m/z* value for starting and ending the integration. Since the identity, i.e., their elemental composition, of the desired peaks is known, their *m/z* profiles can be calculated with freely available online software at the desired instrument resolution.

Figure 5 displays the simulated appearance of the profiles of the main signals in the MALDI analysis of the hemolysate.

The next step to reliably assess the area of the peaks that correspond to the ion species of interest (i.e., the MH+ species of β-Hb and β-Hb-SSG) is determining, for the mass-intensity profile of each spectrum, the correct value of the baseline. The profile spectra were not baseline-subtracted or smoothed with the FlexAnalysis™ software prior to export, but were only calibrated with the four signals of doubly- and singly-charged α- and β-Hb to avoid introducing a preliminary bias in the signal intensity and peak shape.

For each spectrum, a starting estimator of baseline intensity is the median value of the signal throughout each spectrum. Its intensity can be modulated through a multiplicative parameter for each spectrum that the Operator adjusts to minimize the uncertainty of the quadruplicate measurement, measured as the standard error calculated in the “analysis report” page of the spreadsheet. Since the same horizontal (constant) baseline value is applied to all signals of the spectrum, and especially to the intense β-Hb and to the much less intense β-Hb-SSG ions, the chance that the Operator unwillingly biases the measurement towards a value is actually minimized. An excessive, or insufficient correction of the baseline has the same (detrimental) effect on the “large” and the “small” peak. In addition, since the four instrumental replicates (spots) actually represent the same sample, measurements should converge to the same value.

### 2.5. Analytical Figures-of-Merit and Example Results from Studies

The improved procedure has been employed in the measurement of more than 300 blood samples that were obtained in clinical studies with different aims. Most results have been published in individual articles, or will be published soon.

This method is able to detect glutathionyl-hemoglobin at minimum fractions in the range of 0.5% and to quantify the ratio to the unmodified β-chain with a median standard error of ±0.2% (IQR 0.2%–0.3%) in most samples. 

Some articles report HbSSG% in samples as the ratio of HbSSG to the sum of unmodified β-chain and of glycated β-chain. Thus, the contemporary measurement of glycated hemoglobin is necessary. In the more than 350 samples analyzed for this work, glycated hemoglobin could be measured at a median value of 5% (IQR 4.3%–5.9%; min 2.6%–max 13.6%) with a median precision of 0.5% (min 0.1%–max 2.2%). This figure refers to low physiological levels of glycated hemoglobin as observed in the healthy non-diabetic population and, from the analytical point of view, to the worst case for accuracy and precision.

With the use of the estimated uncertainties for HbSSG and gycated hemoglobin, the calculation of the extended uncertainty of the measurement affords a median value of 6.1%, expressed as the CV% of the HbSSG/(β-Hb + Glc-β-Hb) ratio.

Inter-day reproducibility was assessed for 23 samples, analyzed in duplicate consecutive sessions one week apart, from hemolysates freshly prepared from sub-samples (50-microL RBC concentrates) of the same RBC preparation. HbSSG% values ranged from 6% to 28% (median 13%). Median difference of the values obtained in the two sessions was 0.7% (IQR −0.3%–1.8%; min-max −3.8%–4.7%). In the same samples, measurement of glycated hemoglobin yielded values ranging from 4% to 8% (median 5%). Median difference of the values obtained in the two sessions was 0.7% (IQR −0%–1.4%; min-max −1.2%–2.8%). Figure 6 shows the plot of the two replicate measurements.

Samples derive from four main lines of study. The first group (69/334 samples; 92 measurements) are subjects who dwelt at the Concordia (Dome-C) Antarctica scientific station during 2012 [19,20]. The second group (32/334 samples; 32 measurements) are trained amateur breath-hold divers who participate in physiological studies of cardiovascular and pulmonary fitness in Y-40 “The Deep Joy” swimming pool (Montegrotto Terme, PD Italy) [21]. The third group (209/334 samples; 221 measurements) are from hospital patients with COVID-19 and other non-infective conditions of respiratory impairment and a group of age-matched control subjects [22,23]. The fourth group (24/334 samples, 24 measurements) are patients with several kidney impairments who were undergoing kidney imaging with Technetium-99m radionuclide tracer.

Representative measurements of the four sample groups are reported in Table 2.

Samples were measured in 11 sample batches over one year. The number of samples in each batch and the representative values measured in each are reported in Table 3. Batches were organized on an “as needed” basis, with samples deriving from one or from several studies. This organization alone explains the large variation of HbSSG levels measured in some batches and the homogeneity of values measured in others.

As reasonable, considering the variety of the studies that involve the measurement of glutathionyl hemoglobin, the results from so different samples cover a wide range of percent HbSSG values, which span from undetectable levels (approx. below 0.5%) to approx. 30%. Median and mean values of glutathionyl hemoglobin in the samples are 6.5% and 6.8%, as shown in the cumulative distribution plot of Figure 7. 

Figure 8 shows a plot of the Standard Error of the individual determinations (quadruplicate measurement of each sample).

The values of the SE span from ±0.03% to ±1.2%, with a median SE of ±0.2% and median values for the batches between ±0.1% and ±0.5%. The insert of Figure 8 shows a plot of SE vs. HbSSG percent in the samples with a few extreme SE values measured both for very low and for some very high values. While the former corresponds to samples where HbSSG is in fact absent, SE for a few high-level samples does not exceed ±0.8%. The corresponding Coefficient-of-Variation (CV% = SE/HbSSG) has a median value of 3.7% (inter-quartile range 2.0–9.6%).

Reproducibility of the procedure was assessed through the duplicate analysis of samples within the same or among different measurement batches.

An intra-day reproducibility test involved the duplicate measurement of 12 samples within the same batch. Each sample was prepared from a separate 50-microL aliquot fractioned from the same blood specimen and analyzed in the same session. Values of HbSSG% in this group of samples were between 2.2% and 3.6%, with a median of the absolute differences between values of 0.7% (min-max 0.2%–2.0%).

Another test involved 23 samples with a relatively high level (6.5%–23.9%) of HbSSG, in two batches that were prepared from the hemolysate and measured one month apart. Individual measurements in each batch had a median SE of ±0.3% (min-max of ±0.1%–±0.7% for the first batch; of ±0.1%–±0.5% for the second batch). The earlier batch is taken arbitrarily as reference for the measurement of the second one. The median difference between duplicate measurements was −0.4% (IQR of −1.5%–0.3%), however with differences as extreme as ±3.8%, corresponding to CV% of approx. 40%. This apparently high difference was observed in RBC concentrates that had not been prepared for long-term storage and analysis, thus representing a rather extreme case that points to the necessity of a proper sample fractionation and storage.

Sample stability for the measurement of HbSSG had been tested separately in a study on Red Blood Cells concentrates from the Hospital’s Blood Bank. The study highlighted that the levels of HbSSG in the 50 tested donors’ blood was mostly below the measurement threshold of 0.5% (6 subjects had HbSSG levels between 0.5% and 0.8%). Under the preservation conditions of the Blood Bank, there was no further production of HbSSG after 27 and after 42 days of storage, the upper limit for use in transfusion [24].

## 3. Materials and Methods

The Appendix A documents reports the complete working protocol with full details on the individual steps, the operating instructions for the calculation spreadsheet and the calculation spreadsheet itself with a full example of a sample analysis.

### 3.1. Cohorts and Samples

Whole blood samples were obtained within the frame of several different research projects that were authorized over time by the Human Ethic Committee of the participating hospital institutions.

Hematocrit and hemoglobin concentration are measured in each blood sample by the hospital laboratory facility, or in the sample delivered to the analytical laboratory, whenever possible. Samples received fresh by the analytical laboratory are immediately fractioned to obtain sub-samples of serum, plasma and red blood cells, according to the needs of the individual project. Fractioned samples are stored at −20°C or at −80°C until preparation for analysis of the required biochemical parameters.

Operators routinely implement full safety precautions in the fractionation phase of sampling, also taking into account those especially established for the management of blood samples collected during the 2020–2021 pandemic episode. No stricter-than-usual safety measures were established for the manipulation of red blood cells, since the combination of prior sample freezing and extensive dilution for hemolysis are deemed sufficient to clear biological hazard.

### 3.2. Equipment

A Medonic M20 hemocromocytometer (Boule, Spånga, Sweden) was employed to measure hematological parameters in the freshly withdrawn blood samples. Measured parameters were downloaded analogically to a prepared reference chart and uploaded into the spreadsheet for calculation.

An Encyte (Perkin-Elmer, Monza, Italy) spectrophotometric plate reader was employed to measure hemoglobin concentration in 96-well disposable plates. Measurements were performed at 360, 405 and 454 nm. Downloaded absorbance values were transferred to a custom spreadsheet for further elaboration and calculations.

An Autoflex III (Bruker Daltonics, Bremen, Germany) Matrix-Assisted Laser-Desorption Time-of-Flight (MALDI-ToF) mass spectrometer was employed for the final measurements. The instrument is equipped with a 355 nm Smart Beam solid state Nd:YAG UV laser and a 396-well plate sample holder. The instrument was operated according to the manufacturer’s directions in the linear positive ion mode at a calculated resolution of approximately 1000. Pre-run external calibration of the *m/z* scale was accomplished with the four main mass peaks in the spectrum of reagent-grade human hemoglobin. Automated routines for the analysis of a large number of samples were established with the Flex Control™ software. Post-acquisition calibrated mass-intensity binary files were obtained for each analysis, and transferred to custom spreadsheets for further elaboration and calculations.

### 3.3. MALDI Measurement

For each measurement batch, the MALDI matrix was prepared by freshly dissolving a weighted amount of solid sinapinic acid (SA) in the appropriate volume of a spot-prepared mixture of acetonitrile and 0.1% solution of trifluoroacetic acid in water, to a final 0.1 M concentration of SA. The SA was pre-weighted in appropriate amounts to analyze batches of 24, 48 or 96 samples, considering 10 microL of matrix solution per sample, plus a 50 microL excess to facilitate accurate pipetting.

For each measurement batch, one portion of frozen separated RBC from each sample is thawed and 1:100 primary haemolysates are prepared. Next, hemoglobin concentration is measured spectrophotometrically and titrated sample solution (0.5 or 1.0 mL) is prepared by diluting the primary hemolisate at 10 microM. Aliquots of 10 microL of titrated hemolysate are taken into 0.2-mL sample cones and stored at −20°C until measurement, which usually occurs within 24 hours.

For the MALDI-ToF mass spectrometry measurement, 10 microL of freshly prepared matrix solution are added to the sample cone, which is thoroughly mixed in a vortex shaker for 30 sec, and the sample is centrifuged for 60 s. Four 1-microliter portions from the mixture are applied, each one in the center of a different circular spot of a standard 384-sample ground steel target (Bruker MTP384 S/N 03229). The manual preparation of one batch plate takes approx. 45 min for each 24 samples (96 spot depositions). 

Evaporation of the solvents is accomplished in the free air at room temperature, over approx. 15 min, and observing the loss of glitter in the deposited material.

Sample plates are inserted into the mass spectrometer for analysis within 4 hours from preparation. Any excess solvent is removed in the pre-vacuum phase of sample plate loading in the instrument.

For analysis, desorption ionization of sample spots is accomplished by firing the laser beam according to an automated routine and accumulating valid spectra until a suitable signal-to-noise ratio is achieved. Measurement on an individual spot takes approximately 45–60 s. 

After acquisition, MALDI spectra are analyzed off-site. Spectra calibration and annotation is performed with the FlexAnalysis™ software. Binary mass-intensity. txt files are obtained for each analyzed spot and exported to a custom Microsoft Excel® worksheet for further elaboration.

### 3.4. Post-Acquisition Elaboration and Calculation of Glutathionyl-Hemogobin Fraction

All calculations, statistical elaborations and graphics were performed in custom Microsoft Excel® worksheets. A copy of the spreadsheets with written instructions is attached to this article as Appendix A.

Briefly, the four binary mass-intensity files of each blood sample are aligned in the calculation spreadsheet and the spreadsheet normalizes the intensities of the four profiles to that of the alpha-hemoglobin MH^+^ species, which is the most intense signal in the profile spectrum. The main human hemoglobin molecular species in the MALDI-ToF spectrum are identified in the *m/z* axis by comparison with those calculated, on the basis of their atomic composition and charge state, with the use of an on-line tool (https://www.envipat.eawag.ch/index.php; last accession 4 November 2021).

Glutathionyl-hemoglobin is calculated as the ratio of the integrated areas of the single-charged (MH^+^) species in each of the four replicate measurements.

The spreadsheet evaluates the baseline value of each spectrum as the median value of its intensity, and subtracts a multiple of this value from the raw-spectrum signal intensity. As the subtracted baseline intensity value is changed, the spreadsheet integrates in each of the four spectra the peak areas of the main protein species from the baseline-subtracted signals. A module of the spreadsheet calculates the mean and standard error of the four determinations, in order that the operator can change the baseline value of each profile spectrum to minimize the standard error.

## 4. Discussion and Conclusions

A major difficulty to use glutathionyl-hemoglobin measurement by MALDI mass spectrometry as biomarker of systemic oxidative stress in human clinical and epidemiological studies is the lack of sufficient traceable precision by using the standard data processing software of one widely available commercial MALDI-ToF instrument. This difficulty has insofar hampered the diffusion of glutathionyl hemoglobin measurement as a biomarker of systemic oxidative stress. No reference material is commercially available for glutathionyl hemoglobin, since this biomarker is not commonly used in clinical chemistry, thus there is little interest in undertaking analytical quality studies. Ours is the first attempt to optimize analytical steps and to assess the outcome of the measurement. The stability of glutathionyl-hemoglobin in preserved separated red blood cells is itself a conflicting issue that can start being addressed only with the use of an analytical method of which the quantitative performance has been characterized.

Another modified form of hemoglobin, glycated hemoglobin (GlcHb or HbA_1c_), is a widely used clinical biomarker of long-term glycemic control in the follow-up of diabetic subjects. Due to its importance, several methods have been developed for measurement, spanning in complexity from automated colorimetric tests to high-performance liquid chromatography separation that can be traced to a reference method and to reference materials [25].

Improved data quality may encourage expanding the use of glutathionyl-hemoglobin as a biomarker of systemic oxidative stress in several human conditions and diseases [26], to compare it with other biomarkers of the thiol-disulphide metabolome, such as the glutathione-glutathione disulfide redox pair [3,5,27]. Accurate and precise measurement of glutathionyl hemoglobin with the fast MALDI-ToF mass spectrometry technique can help solve still poorly understood issues on the different mechanisms of its generation. In particular, dose-response curves of production and reduction of glutathionyl-hemoglobin in red blood cells challenged by different oxidants, such as *t*-butyl-hydroperoxide in the case of [3], may identify different mechanisms of RBC oxidative stress [2,3,28,29] and demonstrate the efficiency of coping mechanisms, such as lifestyle and nutraceutical interventions.

## Figures and Tables

**Figure 1 molecules-28-00497-f001:**
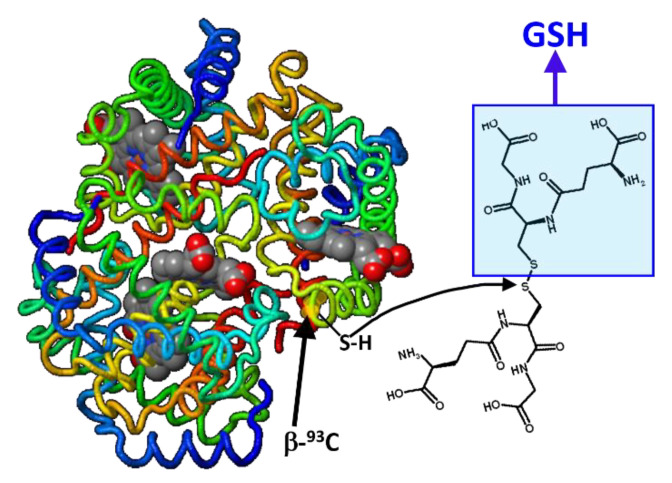
Formal mechanism for the generation of glutathionyl-hemoglobin by thiol exchange of glutathione disulfide with the thiol(ate) group in the side chain of one β-93-Cys residue (h-Hb structure modified from https://proteopedia.org/fgij/fg.htm?mol=3ODQ, accessed on 30 September 2022).

**Figure 2 molecules-28-00497-f002:**
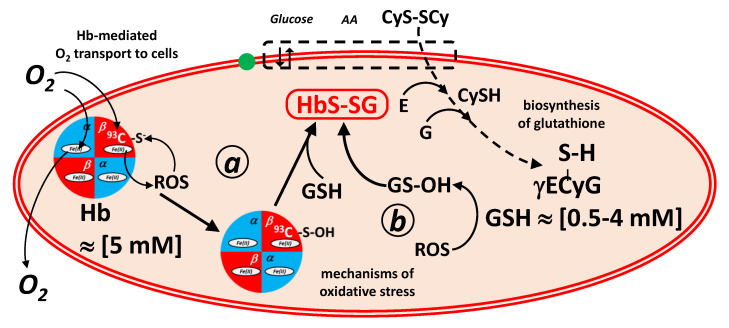
Formal mechanisms of generation and coping of oxidative stress in red blood cells mediated by hemoglobin and glutathione redox buffers. In mechanism ***a***, oxy-hemoglobin generates Reactive Oxygen Species (ROS, such as hydrogen peroxide) that convert by oxidation the ^93^β-Cys residue into the corresponding sulphinic acid. The hemoglobin sulphinic acid reacts with glutathione (GSH; γECyG) to generate glutathionyl-hemoglobin. In mechanism ***b***, ROS oxidize glutathione to glutathione sulphinic acid (GS-OH), which reacts with the ^93^β-Cys residue of hemoglobin, again to generate glutathionyl-hemoglobin. Highlighted are (left) the transport of O_2_ to the cells, mediated by hemoglobin, and (right) the biosynthesis of glutathione in erythrocytes from the constituent amino acids, glutamic acid (E), glycine (G) and cysteine (CySH), which is imported as its disulfide, cystine (CyS-SCy).

**Figure 3 molecules-28-00497-f003:**
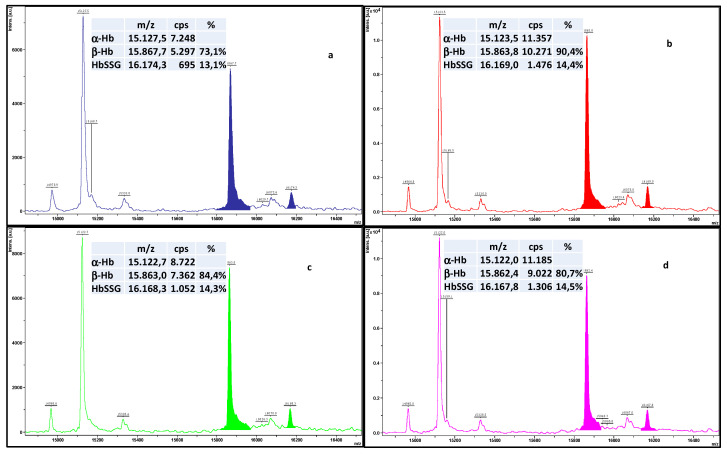
Superimposed traces of four depositions (**a**–**d**) from the same hemolysate-matrix mixture, measured with the automated acquisition method (50 shots summed over the sample spot area). Colour-highlighted areas correspond to the integration by the FlexAnalysis™ software, achieved under the best operating conditions and with optimized post-acquisition parameters. The boxes summarize the main measured peak areas and the relevant derived calculations.

**Figure 4 molecules-28-00497-f004:**
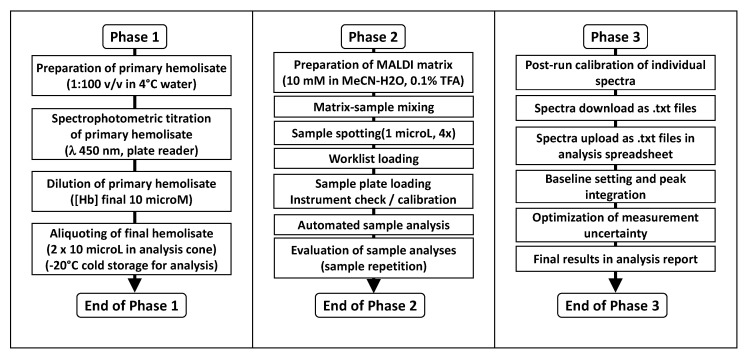
Three-phase analysis flowchart for the measurement of glutathionyl-hemoglobin in human red blood cells. The three phases are organized to allow manual sample preparation, analysis and processing in batches by a single Operator with data reporting over a weekly shift.

**Figure 5 molecules-28-00497-f005:**
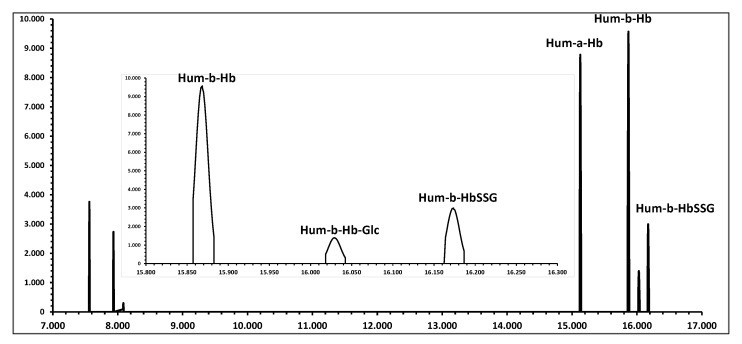
Simulated MALDI spectrum of a human red blood cell hemolysate with typical relative ratios of signals (see Table 1). The insert displays the expanded region of interest for the quantification of the ion signals for the measurement of glutathionyl-hemoglobin, as they appear at a resolution of 1000 DM/M.

**Figure 6 molecules-28-00497-f006:**
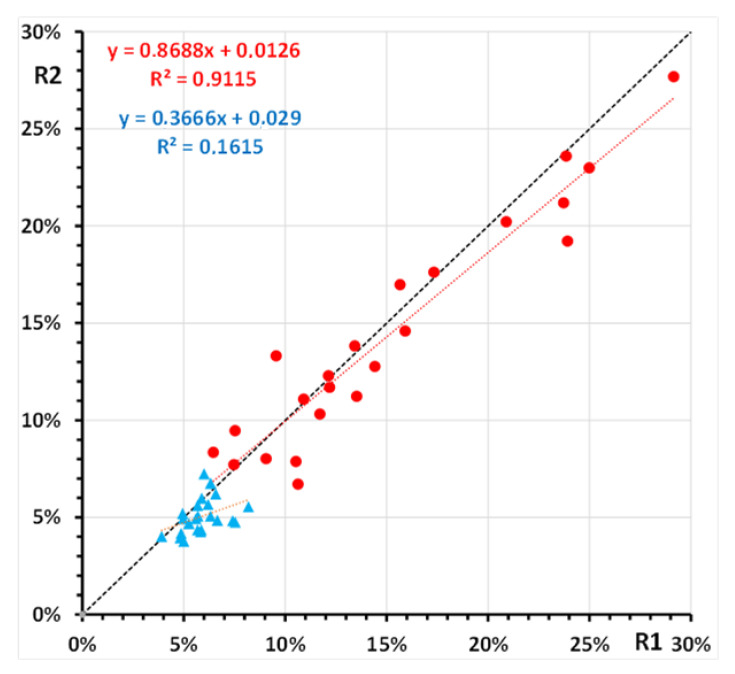
Replicate measurement of glutathionyl-hemoglobin (●) and of glycated hemoglobin (▲) in 23 samples of a human red blood cell hemolysate. Shown are the least-squares regression lines of the second replicate (R2) *vs.* the first replicate (R1) for both hemoglobin forms.

**Figure 7 molecules-28-00497-f007:**
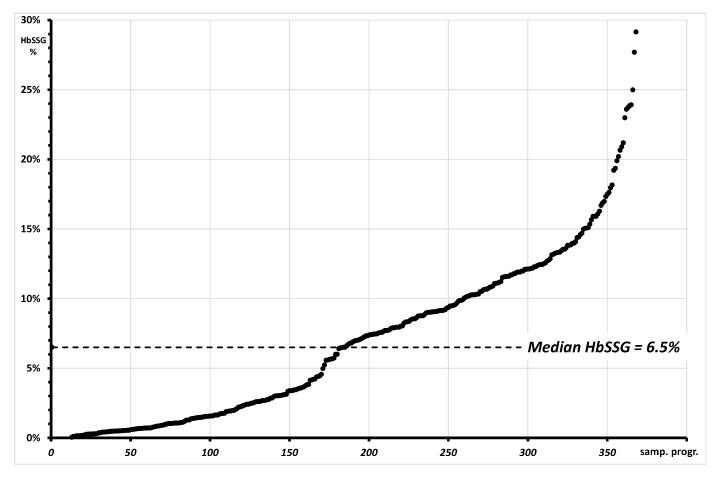
Plot of the cumulative distribution of 368 measurements of glutationyl hemoglobin with values spanning from “not measurable” (approx. <0.1%; 23/368) to approx. 30%. The graph reports as a dotted line the median value of HbSSG in the measured samples.

**Figure 8 molecules-28-00497-f008:**
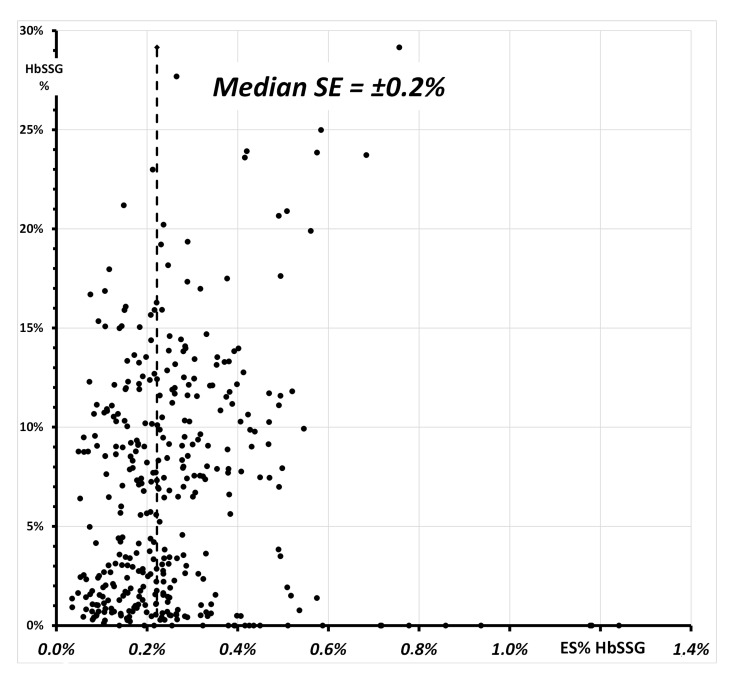
Relationship of the SE% of the 368 measurements of glutationyl hemoglobin (Figure 7) and of the respective value of HbSSG%. The vertical line highlights the median value of the SE in the measured samples.

**Table 1 molecules-28-00497-t001:** Calculated elemental composition and *m/z* value of the main protein species of interest in the MALDI spectra of human red blood cell hemolysates.

Protein ID	Composition ^1^	Charge State	*m/z*
Hum-HbA	C_685_ H_1073_ N_187_ O_194_ S_3_	[MH_2_]^2+^	7563.59
Hum-HbB	C_724_ H_1119_ N_195_ O_201_ S_3_	[MH_2_]^2+^	7934.01
Hum-HbA	C_685_ H_1072_ N_187_ O_194_ S_3_	[MH]^+^	15,127.17
Hum-HbB	C_724_ H_1120_ N_195_ O_201_ S_3_	[MH]^+^	15,868.02
Hum-HbB-Glc	C_730_ H_1129_ N_195_ O_206_ S_3_	[MH]^+^	16,030.16
Hum-HbBSSG	C_734_ H_1135_ N_198_ O_207_ S_4_	[MH]^+^	16,174.33

^1^ All *m/z* values and molecular isotope cluster profiles (see Figure 4) have been calculated for a value of resolution of 1000 (M/DM) by the online free software (https://www.envipat.eawag.ch/index.php; last accession 24 April 2022).

**Table 2 molecules-28-00497-t002:** Summary of the results from the measurement of glutathionyl hemoglobin in RBCs of subjects belonging to different study groups.

Group	n° Samples	Min	Max	Median	ES% ^1^	Notes
**1–Antarctica**	69	0.7%	16.9%	7.2%	0.2%	[12,13] ^2^
**2–Breath-hold divers**	32	5.6%	16.7%	9.8%	0.3%	[14] ^2^
**3–ARSD**	26	0.0%	16.9%	2.5%	0.2%	[15] ^3^
**4–COVID-19**	165	0.0%	15.9%	1.4%	0.2%	[16] ^3^
**5–COPD**	14	3.4%	12.6%	9.2%	0.2%	[15] ^3^
**6–Age-matched controls for groups 3, 4, 5**	9	2.4%	12.0%	5.9%	0.2%	[14] ^2^
**7–Nephrotic patients**	24	0.0%	1.5%	0.0%	0.5%	Unpubl.

ARSD: Acute Respiratory Stress Disease; Chronic Obstructive Pulmonary Distress; ^1^ median value of the Standard Error of the individual samples in the batch. ^2^ Other biomarkers; HbSSG% not reported. ^3^ HbSSG% reported among other biomarkers.

**Table 3 molecules-28-00497-t003:** Summary of the results from the measurement of glutathionyl hemoglobin in 11 batches of RBCs.

Batch	n° Samples	Min	Max	Median	ES% ^1^	Notes
**1**	77	0.7%	16.9%	7.2%	0.2%	
**2**	54	5.6%	16.7%	9.8%	0.3%	
**3**	56	6.5%	29.2%	13.4%	0.3%	12 samples of batch 3 re-analyzed in batch 8
**4**	24	0.0%	1.5%	0.0%	0.5%	
**5**	24	0.3%	2.9%	0.7%	0.1%	
**6**	24	1.4%	3.8%	3.0%	0.2%	12 samples in duplicate
**7**	19	0.5%	3.1%	1.5%	0.2%	
**8**	24	6.5%	27.7%	12.5%	0.3%	12 samples of batch 3 re-analyzed in batch 8
**9**	24	0.0%	3.6%	1.4%	0.2%	
**10**	25	0.0%	12.9%	4.2%	0.2%	
**11**	17	0.0%	2.4%	0.7%	0.2%	

^1^ median value of the Standard Error of the individual samples in the batch.

## Data Availability

Raw and processed data for the samples of this study are stored in digital form by the Authors. A calculation example is supplied in the Appendix A.

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
