# Peer review of "Enhanced-Precision Measurement of Glutathionyl Hemoglobin by MALDI-ToF MS"

_molecules, 2023, doi:10.3390/molecules28020497_

Round 1

Reviewer 1 Report

The present manuscript deals the highly precised measurement of glutathionyl hemoglobin by MALDI-ToF MS. The work was introduced well and the objective is explained clearly in the introduction part. The methods adopted was found to be appropriate. The manuscript also explains the merits and demerits of the existing methods and how the present method is having advantage over the existing methods. I think the results and discussion part is too lengthy and too many details are given, this may be shorten. The images and tables are presented well and discussed well. All the cited references are up to date and appropriate. The manuscript also describes the advantages and disadvantages of the existing methods. Overall, the manuscript may be interested to the readers and the merit is excellent.

Author Response

Reviewer 1.

The present manuscript deals the highly precised measurement of glutathionyl hemoglobin by MALDI-ToF MS. The work was introduced well and the objective is explained clearly in the introduction part. The methods adopted was found to be appropriate. The manuscript also explains the merits and demerits of the existing methods and how the present method is having advantage over the existing methods. I think the results and discussion part is too lengthy and too many details are given, this may be shorten. The images and tables are presented well and discussed well. All the cited references are up to date and appropriate. The manuscript also describes the advantages and disadvantages of the existing methods. Overall, the manuscript may be interested to the readers and the merit is excellent.

Reviewer 1. Authors’ response

Dear Reviewer 1, dear Editor,

Thank you for kind and competent reading of our draft. We took your advice and shortened the text (also the other Reviewer noticed this flaw).

Kind regards

FMR on behalf of the Authors Team

Reviewer 2 Report

Federico Maria Rubino and, colleagues report some important methodology, analysis, and observations for measurement of glutathionyl hemoglobin 2 by MALDI-ToF MS ‘. Their work provides very descriptive details about the improved method and how it could be very useful and applicable to study human oxidative stress in a variety of pathological and living conditions and how it could be exploited  in clinical setup as a biomarker. In general, the paper is well written, and methods are well described, and supportive supplementary materials are provided. I consider this is useful contribution in its field, and acceptable for publication with following minor revisions.

1.      Introduction last para, line from 102-105 author described using operator-controlled tools- “A breakthrough in this development was choosing not to use the vendor-supplied spectrum integration tools, and to develop an original, specific spreadsheet that performs the same task with a totally operator-controlled calculation tool”. But in results section, line from 145-147, they described about the optimization with “This optimization task was necessary since, due to the large number of samples that need to be measured in clinically oriented  studies, measurements should be made operator-independent and unbiased”. As clear from the sample preparation section-it was operator based, analysis was not.

2.     Why two calibration, external and internal was necessary, how they were compatible and why it was done specifically when authors were not using vendor-supplied spectrum integration tools.  As described in line 159-166. “To implement automated spectra processing with the FlexControl™ software, routine external calibration of the m/z scale is accomplished with the manufacturer’s custom protein mixture only on a periodic basis”.

3.     Line 201-211- I think I don’t agree fully with the authors about the errors in coding, labeling and mixing etc. for the samples for the analysis, to my best of knowledge most clinical diagnostic laboratories follow very  strict GLP, and CLIA certified, and use automated methods for coding/ labeling mixing or aliquoting  the samples which include high quality consumables. Few cases of mishap can occur under very strict regulation in any facility.

4.     Results sections-line-257-267 is too descriptive. Most of the results section for that matter is too descriptive, can be shortened.

5.     Discussion and conclusion part has some repetitive lines already mentioned in results and introduction.

Author Response

Reviewer 2.

Federico Maria Rubino and, colleagues report some important methodology, analysis, and observations for measurement of glutathionyl hemoglobin 2 by MALDI-ToF MS ‘. Their work provides very descriptive details about the improved method and how it could be very useful and applicable to study human oxidative stress in a variety of pathological and living conditions and how it could be exploited in clinical setup as a biomarker. In general, the paper is well written, and methods are well described, and supportive supplementary materials are provided. I consider this is useful contribution in its field, and acceptable for publication with following minor revisions.

Reviewer 2. Authors’ response

Dear Reviewer 2, dear Editor,

Thank you for kind and competent reading of our draft. We understand the points that you raise and we introduce some clarifications here and in the revised text.

  1. Introduction last para, line from 102-105 author described using operator-controlled tools- “A breakthrough in this development was choosing not to use the vendor-supplied spectrum integration tools, and to develop an original, specific spreadsheet that performs the same task with a totally operator-controlled calculation tool”. But in results section, line from 145-147, they described about the optimization with “This optimization task was necessary since, due to the large number of samples that need to be measured in clinically oriented studies, measurements should be made operator-independent and unbiased”. As clear from the sample preparation section-it was operator based, analysis was not.

There might be a little ambiguity from our part in the use of the terms “operator-independent” and “operator-controlled”, and we revise the text to eliminate the incongruence. The spreadsheet is much easier to handle by any (“operator-independent”) operator: we tested its ease of operation by having one of the co-authors see one procedure and then elaborate successfully a few spectra on her own (“operator-independent”). We organized the spreadsheet in order that the operator cannot freely skew the measurement (“unbiased”), but only strive for the highest possible precision in the evaluation of area ratio of the four replicates. This is obtained by modifying only one parameter that increases or decreases the constant baseline (“operator-controlled”).

  1. Why two calibration, external and internal was necessary, how they were compatible and why it was done specifically when authors were not using vendor-supplied spectrum integration tools. As described in line 159-166. “To implement automated spectra processing with the FlexControl™ software, routine external calibration of the m/z scale is accomplished with the manufacturer’s custom protein mixture only on a periodic basis”.

In fact, this information is confusing and we strike the sentence altogether. Just for you to know, this procedure is part of the internal organization of the facility that hosts the shared instrument. Calibration with the standard protein mix is part of the routine procedure and can be performed either by the staff or by one of the (few) authorized direct users. Calibrations are saved over time to allow the vendor’s service to perform their checks in case of problems. Sometimes, especially in the past years, there was some nuisance among users who used the instrument in different modes for different purposes (protein digests, low mass range; proteins, high mass range; reflectron mode or reflectron-off mode; tandem-MS). We therefore agreed that the user should leave the instrument calibrated in the used mode with the vendor’s standard mix. Deposition of the standard mix, of course, needs a separate spot and this disturbs our multi-sample routine. Time ago, even before we introduced our integration tool, we agreed with the staff that we calibrate the reflectron-off protein range with our hemoglobin peaks by using our separate hemoglobin reference mass table, which is appended to the standard one but is stored and retrieved from our dedicated directory, and share the calibration spectrum for instrument performance checks.

  1. Line 201-211- I think I don’t agree fully with the authors about the errors in coding, labeling and mixing etc. for the samples for the analysis, to my best of knowledge most clinical diagnostic laboratories follow very strict GLP, and CLIA certified, and use automated methods for coding/ labeling mixing or aliquoting the samples which include high quality consumables. Few cases of mishap can occur under very strict regulation in any facility.

We fully agree. However, we are not a clinical diagnostic laboratory that operates with automated handling devices under GLP, but a small research laboratory. We perform this measurement on batches of a few to several tens of samples that come in different forms (whole raw blood tubes to fractionate, separated RBCs; fresh samples; stored samples). Since all procedures are manual, coding and keeping the codes right throughout the multiple passages is a critical step of our work. We have put your caveat in the sentence to improve understanding.

  1. Results sections-line-257-267 is too descriptive. Most of the results section for that matter is too descriptive, can be shortened.
  2. Discussion and conclusion part has some repetitive lines already mentioned in results and introduction.

Thank you for this suggestion. We simplify and shrink both sections.

Kind regards

FMR on behalf of the Authors Team